# In-silico characterization of deleterious non-synonymous SNPs in the human *S1PR1* gene reveals structural instability and altered ligand affinity

Sangram Biswas[1], Dipankar Sardar[1], Md. Arju Hossain[2], Soharth Hasnat[3], Arif Hossain Ramjan[4], Ummah Kulsum Nazifa[1], Fatema-Tuz Zohora[1], Ishrat Jahan Esha[1], Chandrika Mondal[5], Abdul Barik[1], M. Nazmul Hoque[3]*

1 Department of Biotechnology, Bangladesh Agricultural University, Mymensingh, Bangladesh,
2 Department of Biochemistry and Biotechnology, Khwaja Yunus Ali University, Sirajganj, Bangladesh,
3 Molecular Biology and Bioinformatics Laboratory, Department of Gynecology, Obstetrics and Reproductive Health, Gazipur Agricultural University, Gazipur, Bangladesh, 4 Department of Pharmacy, Mawlana Bhashani Science and Technology University, Tangail, Bangladesh, 5 Department of Fisheries Biology and Genetics, Bangladesh Agricultural University, Mymensingh, Bangladesh

* nazmul90@gau.edu.bd

## Abstract

S1PR1 is a G protein-coupled receptor that plays a key role in regulating lymphocyte trafficking, immune response, cardiovascular system function, cell proliferation and survival, tumor angiogenesis, and metastasis. It is also recognized as a pharmacotherapeutic target for the treatment of autoimmune diseases like relapsing multiple sclerosis and ulcerative colitis. This study aimed to identify deleterious non-synonymous single nucleotide polymorphisms (nsSNPs) in the S1PR1 gene that may impact its functional properties and pharmacotherapeutic responses though in-silico approaches. A total of 3,259 SNPs were identified in the human S1PR1 gene, with 6.51% being non-synonymous (nsSNPs). Functional predictions from eight computational tools prioritized 25 deleterious variants. Further structural and evolutionary evaluation highlighted R120P, F125S, C184Y, Y198C, and L275P as the most damaging nsSNPs. These mutations were found to cluster within the seven-transmembrane (7-TM) domain (residues 46–322), directly affecting receptor stability and signaling. Structural modeling revealed disrupted hydrogen bonds, void formations, and loss of critical disulfide bonding (C184Y), severely compromising folding. Conservation analysis confirmed R120P, F125S, and C184Y as highly conserved (score 9), underscoring their functional importance. Molecular docking and dynamics simulations showed that R120P and F125S weaken binding affinity for natural agonist sphingosine-1-phosphate (S1P) and FTY720P, while antagonist W146 retained strong binding. Our analysis further revealed significant changes in binding interactions and protein-ligand complex stability under simulated physiological conditions. Collectively, these findings identified high-risk nsSNPs in S1PR1 gene

**Data availability statement:** All data and supplementary information supporting the findings of this in-silico study are included within the article.

**Funding:** The author(s) received no specific funding for this work.

**Competing interests:** The authors declare that they have no competing interests.

with potential structural and functional implications, particularly in diseases involving impaired receptor signaling. These findings enhanced our understanding of how specific nsSNPs can influence disease susceptibility, drug response, and receptor function, paving the way for precision medicine approaches in treating autoimmune and inflammatory disorders.

## Introduction

Human genetics research aims to understand how genetic variations influence disease risk. However, identifying pathogenic variants among millions of neutral single-nucleotides, structural, and copy number variants remains a significant challenge [1,2]. Genome sequencing has revealed mutations associated with both Mendelian and common disorders, such as autism, yet prioritizing variants for their relevance to disease remains a major challenge [3,4]. Variant prioritization relies on two primary approaches: statistical association, which connects variants to diseases through case-control studies, and functional evaluation, which assesses single-nucleotide variants based on conservation, population frequency, and predicted functional impact [5]. While statistical methods are limited for rare disorders, functional prioritization has been highly effective in identifying causal mutations. Among the many genetic variants found in the human genome, single nucleotide polymorphisms (SNPs), which are defined as single nucleotide substitutions at specific positions, are the most prevalent [6–8]. Their frequency is around 200–300 base pairs, and they account for 90% of the variance in the genome [9,10]. Non-synonymous SNPs (nsSNPs), particularly missense SNPs that cause amino acid alterations in the encoded protein, are of special interest among SNPs. The structure, function, stability, hydrophobicity, charge, geometry, translation, and inter- or intra-protein interactions can all be changed by these modifications, which can lead to phenotypic changes and disease processes [11,12]. In fact, numerous studies have found that nsSNPs account for at least 50% of the variants associated with hereditary genetic diseases [13]. Consequently, SNP association studies help characterize phenotypes and provide information about the development of drugs and treatments for disorders involving particular genetic variations. However, S1PR1 has a significant number of SNPs, it would be expensive and time-consuming to conduct laboratory experiments on the functional implications of these SNPs. Thus, conducting a computational screening of SNPs is an essential step to filter out potentially harmful variants before proceeding with experimental mutation analysis.

S1PR1 is a member of the G-protein-coupled receptor (GPCR) family and serves as a receptor for the bioactive lipid sphingosine-1-phosphate (S1P) [14]. The S1PR1 protein is primarily involved in regulating processes such as vascular development, endothelial integrity, immune cell trafficking, and vascular permeability [15–17]. The gene's coding sequence is spread across multiple exons, contributing to a functional receptor protein. Notably, the S1PR1 receptor is a key mediator in the S1P signaling pathway, influencing cell proliferation, migration, and survival [18]. Abnormal

regulation of S1PR1 activity has been implicated in various clinical conditions, including cancer, autoimmune disorders, cardiovascular diseases, and neurological disorders [19, 20]. Extensive research has been conducted on S1PR1 dysregulation concerning cancer and immune diseases [21]. In cancer, aberrant S1PR1 signaling drives tumor proliferation, angiogenesis, metastatic progression, and chemoresistance, often by modulating the tumor microenvironment and enhancing survival pathways [22]. S1PR1 is involved in lymphocyte trafficking in immunological diseases like multiple sclerosis, and its dysregulation results in pathological inflammation and autoimmunity [17]. Given the importance of S1PR1 in maintaining physiological homeostasis, understanding the impact of genetic variations, particularly nsSNPs, in this gene thus is crucial for elucidating its functional consequences and potential role in disease.

The *S1PR1* gene, positioned on chromosome 1p21.2, encodes a 382-amino-acid-long protein that features seven transmembrane helices, distinctive to the GPCR family [23]. This study investigated 212 nsSNPs in the S1PR1 protein, focusing on their structural and functional impacts using bioinformatics tools and molecular dynamic simulations. High-risk nsSNPs were identified and analyzed for their effects on protein stability, flexibility, and conformation. While S1PR1's role in tumor suppression is well-known, previous studies have not comprehensively addressed the specific impacts of nsSNPs on its structure and function, nor have they integrated computational and experimental approaches. Addressing this research gap, our findings highlight how deleterious/pathogenic nsSNPs impair S1PR1 receptor function, contributing to disease predisposition and therapeutic resistance, with implications for personalized medicine.

## Materials and methods

### Ethical statement

This computational study did not involve human participants and, as such, ethical approval was not required. The authors further confirm that this manuscript, submitted to PLOS ONE, has been prepared in full compliance with responsible research practices and the established guidelines of publication ethics.

### Study outline

The overall workflow employed in this study is illustrated in **S1 Fig**. This research incorporates multiple computational and simulation-based methods to investigate the functional and structural impacts of nsSNPs in the human *S1PR1* gene.

### Retrieval of SNP data

Missense nucleotide polymorphism (nsSNP) data for the human *S1PR1* gene were systematically retrieved from the NCBI dbSNP database [24]. The corresponding canonical protein sequence (UniProt ID: P21453) was obtained from the UniProt database [25]. These retrieved variants were subsequently analyzed for their potential impact on S1PR1 protein structure and function.

### Functional impact assessment

To comprehensively evaluate the functional consequences of nsSNPs in the *S1PR1* gene, a multi-tool computational approach was employed [26,27]. Variants were classified as "*high-risk*" if they were consistently predicted to be deleterious by all applied in-silico tools, indicating a strong potential to disrupt protein function and contribute to disease pathogenesis. The functional impact of each nsSNP was assessed using a consensus strategy involving eight distinct prediction tools such as SIFT, PolyPhen-2, Meta-SNP, PhD-SNPg, WS-SNPs&GO, PROVEAN, PredictSNP, and SNAP. SIFT (Sorting Intolerant From Tolerant) predicted deleteriousness based on sequence homology and physicochemical properties, with a tolerance index (TI) score ≤0.05 indicating a damaging effect [28]. PROVEAN (Protein Variation Effect Analyzer) used an alignment-based scoring system to assess the functional impact of amino acid substitutions [29]. PolyPhen-2 (Polymorphism Phenotyping v2) estimated the effects of mutations based on sequence conservation and structural parameters [30]. WS-SNPs&GO integrated protein sequence data, gene ontology (GO) annotations, and evolutionary

information to classify variants as disease-related or neutral [31]. PhD-SNPg employed a support vector machine (SVM) classifier to predict pathogenic nsSNPs [32], while Meta-SNP acted as a meta-predictor, combining the outputs of multiple tools for enhanced accuracy [33]. PredictSNP [34] and SNAP [35] offered consensus predictions by aggregating results from various algorithms to evaluate the functional significance of nsSNPs. This integrative approach provided a robust framework for identifying potentially deleterious variants in *S1PR1* and exploring their possible roles in disease development.

### Structural stability prediction

Eight different computational tools namely MUpro, I-Mutant2.0, NetSurfP-3, DUET, mCSM, SDM, HOPE, and Mutation3D were employed to evaluate the structural impact of nsSNPs on the S1PR1 protein. MUpro (https://www.ics.uci.edu/~baldig/mutation.html) utilized support vector machines and neural networks to predict protein stability changes upon point mutations, achieving approximately 84% prediction accuracy [36]. It provided a confidence score ranging from −1–1, where negative values indicate decreased stability. I-Mutant2.0 (https://folding.biofold.org/i-mutant/i-mutant2.0.html) assessed changes in protein stability resulting from single-point mutations based on either sequence or structural data [37]. NetSurfP-3.0 (https://services.healthtech.dtu.dk/services/NetSurfP-3.0/) was used to predict local structural features, such as solvent accessibility, based on the input sequence, providing insight into the surface exposure of specific residues [38]. DUET, mCSM, and SDM (http://biosig.unimelb.edu.au/duet/) analyzed the effects of nsSNPs by integrating structural and evolutionary information, using the native protein sequence and the mutation site as inputs [39]. The HOPE server (http://www.cmbi.ru.nl/hope/input/) predicted structural consequences of mutations by aggregating data from tertiary structures, sequence annotations, homology models from Distributed Annotation System (DAS) servers, and the UniProt database [25]. Mutation3D (http://mutation3d.org) identified spatial clusters of amino acid substitutions in protein tertiary structures, often associated with cancer driver mutations [40]. This comprehensive structural analysis provided valuable insights into the conformational and stability changes potentially caused by nsSNPs in *S1PR1* gene.

### Domain identification

The functional domain(s) of the human S1PR1 protein were identified using the Pfam database (http://pfam.xfam.org), a widely utilized resource based on UniProt Reference Proteomes. Pfam comprises a curated collection of protein families, each defined by two types of multiple sequence alignments and a profile hidden Markov model (HMM) [41]. These HMMs, constructed from alignments of representative sequences within each family, served as statistical models to detect and annotate conserved domains across related proteins. The Pfam interface provided both graphical representations and interactive access to protein family data, facilitating the accurate identification of domain architecture in S1PR1 protein [42].

### Evolutionary conservation analysis

To explore the evolutionary relationships of the S1PR1 protein, a phylogenetic tree was constructed using MEGA X software [43]. Nine closely related homologs were retrieved via BLASTp search against the NCBI protein database (https://blast.ncbi.nlm.nih.gov/Blast.cgi?PAGE=Proteins) [43]. The phylogenetic tree was generated using the maximum likelihood method with 1000 bootstrap replicates, and was subsequently visualized and annotated using the Iroki webserver (https://www.iroki.net) [44]. To assess residue-level evolutionary conservation, the ConSurf server (https://consurf.tau.ac.il) was employed, which estimates conservation scores based on phylogenetic relationships among homologous sequences [45]. Conservation scores were calculated using the Bayesian method. Particular emphasis was placed on high-risk nsSNPs located within highly conserved regions, as these variants are more likely to affect protein function and were prioritized for downstream analysis.

## Homology modeling, validation, and secondary structure analysis

Homology modeling of the S1PR1 protein was performed using the SWISS-MODEL server (https://swissmodel.expasy.org) [46]. The FASTA sequence of the S1PR1 protein was provided as input, and a suitable template structure (SMTL ID: 7VIF) showing 100% sequence identity with the target sequence was selected for model generation. The quality of the predicted 3D structure was assessed using ERRAT and PROCHECK. ERRAT (https://saves.mbi.ucla.edu/) evaluated the reliability of the model by identifying regions with abnormal non-bonded atomic interactions, a critical criterion for validating structures derived from X-ray crystallographic data [47]. PROCHECK (https://servicesn.mbi.ucla.edu/PROCHECK/) was used to examine the stereochemical quality of the model by assessing backbone dihedral angles and overall geometric conformance of residues [27]. Additionally, STRIDE (https://webclu.bio.wzw.tum.de/stride/) was employed to analyze the secondary structure elements of the modeled protein, identifying features such as alpha helices, beta strands, and beta hairpins based on hydrogen bonding patterns and atomic geometry [48].

## Interatomic interaction prediction

The DynaMut2 server (https://biosig.lab.uq.edu.au/dynamut2/) was employed to predict changes in interatomic interactions induced by missense mutations [49]. The wild-type protein structure in PDB format, along with the corresponding list of point mutations, was provided as input. DynaMut2 combined optimized graph-based signatures with normal mode analysis to evaluate the impact of mutations on protein stability and conformational dynamics. In addition to estimating stability changes, the tool predicted alterations in protein flexibility and interatomic contacts, offering detailed insights into how missense variants may influence protein structure, function, and potentially contribute to disease pathogenesis [49].

## Molecular docking analysis and binding site prediction

Molecular docking studies were performed to investigate the binding affinities and interaction profiles of selected ligands *viz*. S1P, FTY720P, and W146 with the S1PR1 receptor. The selection of these ligands was based on their well-characterized pharmacological properties and functional relevance in modulating S1PR1 activity [50]. Sphingosine-1-phosphate (S1P), the endogenous agonist, established the physiological binding reference [51], while FTY720P, a synthetic immunomodulatory analogue, demonstrated prolonged receptor internalization and distinct agonistic effects [52]. In contrast, W146, a selective antagonist of S1P, was included to represent the receptor's inactive conformation, thereby offering a comparative perspective to delineate structural determinants underlying receptor activation and inhibition [53]. Ligand structures were retrieved from the PubChem database (https://pubchem.ncbi.nlm.nih.gov) [53]. The R120P and F125S mutants were selected for docking analysis based on two criteria such as (i) HOPE server predictions identified these residues within the ligand-binding pocket, and (ii) previous studies reported their involvement in agonist binding and receptor activation [54]. Binding pocket identification was carried out using CASTp 3.0 (http://sts.bioe.uic.edu/castp/calculation.html), which analyzed topological and geometric properties of surface-accessible cavities using the protein's PDB structure [55]. The identified binding sites were consistent with previous crystallographic and in-silico findings involving S1PR1 ligands [54]. Prior to docking, energy minimization of protein structures and ligands was performed using SwissPDB Viewer (LINK/REF) and Avogadro, respectively [56]. Molecular docking was executed using AutoDock Vina, integrated within the PyRx platform [57], under default parameters. The grid box was defined to encompass the predicted agonist-binding site. Post-docking, the resulting protein-ligand complexes were visualized and analyzed using Biovia Discovery Studio Visualizer (v21.1.0) to examine detailed interatomic interactions and binding orientations [27,58].

## Molecular dynamics simulation

The previously developed homology model of S1PR1 was incorporated, as no confirmed crystallographic structure was available, and two-point mutations F125S and R120P were introduced using the PyMOL mutagenesis tool to study

the impact on ligand binding and dynamics. Ligands S1P (CID: 5283560), W146 (CID: 6857802), and FTY720P (CID: 11452022) were selected based on their known interaction with the S1PR1 receptor. The 3D conformers of these ligands were obtained from the PubChem database and energy minimized using the OPLS4 force field in Schrödinger's LigPrep tool [59]. Based on preliminary functional and structural impact predictions, conservation analysis, and molecular docking results, two nsSNPs, R120P and F125S, were selected for in-depth molecular dynamics simulation (MDS). These mutations demonstrated notable alterations in protein stability and interatomic interactions, identifying them as suitable candidates for subsequent MDS with three ligands such as S1P (CID: 5283560), W146 (CID: 6857802), and FTY720P (CID: 11452022). MDSs were conducted using Desmond v24, developed by Schrödinger LLC [60], to validate the interactions identified through docking analysis. MDS employs Newton's classical laws of motion to calculate atomic positions and velocities over time, producing updated configurations at fine time intervals. This method allows the assessment of ligand-binding behaviors of both wild-type and mutant S1PR1 proteins under near-physiological conditions, offering dynamic insights into the stability and interaction profiles of the ligand–protein complexes [61,62]. The MDS were conducted for 200 ns to ensure adequate sampling of the conformational space and to achieve reliable equilibrium of the protein–ligand complex. Shorter simulations (e.g., ≤ 100 ns) often fail to capture slower backbone and side-chain rearrangements, especially in flexible binding pockets. A 200 ns timescale was therefore selected as it allows sufficient relaxation of the system and stabilization of key non-covalent interactions, which are essential for evaluating the dynamic stability and binding persistence of the complexes [59,60]. The protein-ligand complexes (wild-type S1PR1 bound to S1P, FTY720-P, and W146; and mutant S1PR1 (R120P, F125S) bound to the same ligands) were solvated in a cubic box with TIP3P water molecules [63]. The MDS were conducted using the TIP3P solvent model within an orthorhombic box, employing the OPLS_2005 force field under conditions of 300 K temperature and 1 atm pressure to ensure a realistic simulation environment. Simulations were performed under an NPT ensemble using the Nosé–Hoover thermostat and Martyna–Tobias–Klein barostat. A 2 fs integration time step was used, with hydrogen bond constraints enforced via the SHAKE algorithm. Periodic boundary conditions and the Particle Mesh Ewald (PME) method (10 Å cutoff) were used to handle long-range electrostatic interactions, ensuring stable and reproducible trajectories [26,64,65]. The simulations were executed on an NVIDIA GeForce RTX 4070 GPU, with each 200 ns run requiring approximately 7 hours. Trajectory frames were collected and analyzed using simulation interaction diagrams to evaluate molecular fluctuations and key intermolecular interactions [59,60]. To mimic physiological conditions, each ligand–protein complex was neutralized with counter ions and supplemented with 0.15 M sodium chloride.

### Trajectory analysis

Structural stability and conformational changes of the protein-ligand complexes were evaluated using the Root Mean Square Deviation (RMSD). RMSD values were computed for backbone atoms to assess global stability over the 200 ns simulation [60]. Comparison was made across wild-type and mutated complexes to determine the effect of point mutations on receptor-ligand dynamics. Root Mean Square Fluctuation (RMSF): Per-residue flexibility was analyzed using RMSF calculations to identify changes in local mobility, particularly in regions involved in ligand binding [59,60]. Variations in peak intensity and location were used to infer structural destabilization or rigidity induced by mutations.

### Assessment of protein–ligand interactions

Protein-ligand interaction fractions were computed over the simulation to quantify the frequency of contact between ligands and receptor residues. Residues such as Glu124, Ser128, Val133, and Tyr205 were particularly monitored due to their known role in ligand recognition. Two-dimensional (2D) interaction maps were generated using Maestro (Schrödinger) [66] to visualize key non-covalent interactions, including hydrogen bonds, hydrophobic contacts, and polar interactions [59]. This analysis allowed a detailed comparison of interaction patterns across wild-type and mutated complexes.

## Results

### Distribution and functional impact of human *S1PR1* gene nsSNPs

A total of 3,259 SNPs associated with the *S1PR1* gene were identified, comprising 67.96% in intronic regions, 6.51% as nsSNPs (missense variants), 5.16% synonymous variants, a small proportion of in-frame deletions (0.06%) and in-frame insertions (0.03%), and other categories (20.28%) (S1 Table). Among these, 212 nsSNPs, which result in amino acid substitutions, were selected for further analysis due to their potential to alter protein structure or function. To comprehensively evaluate the functional impact of nsSNPs on *S1PR1* gene, predictions were obtained from eight different computational tools (S2 Table). Among the 212 nsSNPs analyzed, the SIFT server identified 77 variants with potential functional impacts on the S1PR1 protein. To further validate and refine these predictions, multiple computational tools were employed (S2 Table), among which PolyPhen-2 classified 86 nsSNPs as 'probably damaging' and 35 as 'possibly damaging.' Similarly, PROVEAN identified 73 nsSNPs as deleterious, indicating a potential effect on protein function. PhD-SNP predicted 104 nsSNPs as disease-associated, while SNPs & GO, META-SNP, PredictSNP, and SNAP predicted 64, 95, 81, and 63 disease-associated variants, respectively (S2 Table). These variants were considered functionally significant and selected for further detailed analysis.

### Structural impact of deleterious nsSNPs on S1PR1 stability

We subsequently evaluated 25 deleterious nsSNPs (variants) to determine their potential impact on the structural stability of the S1PR1 protein through comprehensive computational analysis of seven different tools (S3 Table). MUpro predicted a reduction in protein stability for all variants except four (A127D, S131F, G122R, and D91A). Similarly, using I-Mutant2.0, 19 of the mutations were predicted to reduce protein stability, whereas six variants (T193P, N307D, S192P, S131F, L212P, and A300V) were associated with enhanced stability. Significant alterations in solvent accessibility were observed between wild-type and mutant residues through NetSurfP 3.0, with the T193P and C184Y mutations causing transitions from buried to exposed residue states. DUET predicted increased stability for A300V, while mCSM and SDM classified most mutations as destabilizing, except for D91A, A300V, and S131F. Project HOPE analysis revealed structural changes associated with amino acid substitutions: 12 variants introduced larger residues, six altered residue charges, and seven increased hydrophobicity. Additionally, seven mutations (T193P, R120P, Y198C, Y81C, S192P, S131F, and D91A) disrupted hydrogen bonding in the protein core, potentially impairing proper folding. Several mutations (e.g., R120P, Y198C, Y81C, I224T/S, L275P, F125S, D91A) were predicted to generate internal voids in the protein structure. The C184Y mutation, involved in critical disulfide bonding, was predicted to severely compromise structural integrity. Furthermore, the R120P mutation disrupted a key salt bridge and an α-helix, significantly impairing the receptor's ability to bind sphingosine-1-phosphate. Integrating results from all predictive tools, eight variants (R120P, Y198C, Y81C, I224T, I224S, L275P, F125S, and C184Y) were identified as having the most deleterious effects on the structural stability of S1PR1 and were selected for downstream analysis (S3 Table).

### Identification of deleterious and covered nsSNPs, and domains in S1PR1 protein

The mutation3D server analysis revealed several non-synonymous SNPs (nsSNPs) within the S1PR1 protein structure, as presented in Fig 1. Of these, R120P, Y198C, L275P, F125S, and C184Y were predicted to be structurally and functionally high-risk mutations, and are highlighted in red to indicate their potential deleterious impact on protein stability and function. In contrast, Y81C, I224T, and I224S were classified as covered mutations (shown in blue), indicating variants consistently identified across multiple databases but not predicted to induce notable structural or functional changes in the protein (Fig 1). A characteristic seven-transmembrane (7-TM) domain was identified in the S1PR1 protein, spanning amino acid residues 46–322. Notably, all five selected nsSNPs (R120P, Y198C, L275P, F125S, and C184Y) were mapped

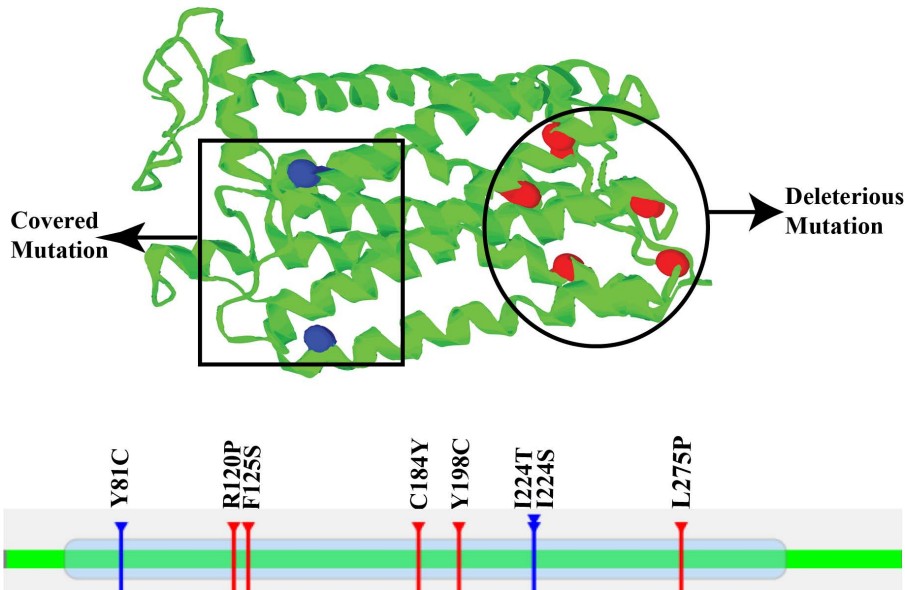

**Fig 1. Three-dimensional (3D) representation of mutations in human S1PR1 protein generated by the Mutation 3D server.** Potentially deleterious nsSNP mutations are highlighted with red markers, whereas blue markers indicate mutations that are structurally mapped but not predicted to be harmful.

within this critical 7-TM domain, suggesting that these mutations may directly influence the structural stability and functional properties of the receptor (**Fig 1**).

## S1PR1 protein conservation and evolutionary dynamics

The sequence conservation analysis of the S1PR1 protein revealed a distinct pattern of highly conserved residues, the majority of which were predicted to be buried (b) by the neural network algorithm. Functional residues (f), characterized by being both highly conserved and exposed, were distributed across the sequence, suggesting their involvement in key functional sites. Structural residues (s), identified as highly conserved and buried, indicated critical roles in maintaining protein stability. Exposed residues (e) were observed in variable regions, potentially influencing ligand interactions. The presence of highly conserved segments underscored the evolutionary importance of S1PR1, implying that mutations within these regions could have severe functional consequences. According to ConSurf analysis, three of the five nsSNPs (R120P, F125S, and C184Y) were located in highly conserved residues with a conservation score of 9. Y198C was classified as moderately conserved with a score of 8, while L275P exhibited an average conservation score of 6. Furthermore, the analysis predicted R120P to be an exposed functional residue, whereas Y198C, L275P, F125S, and C184Y were associated with buried structural residues (**S2 Fig**, **S4 Table**). Overall, R120P, F125S, and C184Y were identified as the variants most likely to disrupt the structural integrity of the S1PR1 protein. Evolutionary phylogenetic analysis of the human S1PR1 protein, along with nine reference S1PR1 proteins from different primate species, revealed a high degree of conservation across primates. Branch lengths and clustering patterns demonstrated strong evolutionary stability, supported by a similarity score of 0.94 (**S3 Fig**). The human (*Homo sapiens*) S1PR1 protein clustered closely with that of *Pan troglodytes*, *Pongo abelii*, and other great apes, consistent with their recent common ancestry. In contrast, more distantly related species, such as *Colobus angolensis palliatus* and *Rhinopithecus roxellana*, occupied divergent branches, indicating earlier evolutionary separation (**S3 Fig**). Collectively, these findings highlighted the functional significance of S1PR1 proteins in primates and suggested that nsSNPs within this gene may have critical biological implications.

## Homology modeling and structural validation of wild-type and mutant S1PR1 proteins

The 3D structure of the wild-type S1PR1 protein (7VIF) and three mutant variants were modeled using the SWISS-MODEL server. Model quality was validated using PROCHECK and ERRAT, with all models satisfying the ERRAT quality threshold. The Ramachandran plots generated by PROCHECK showed that more than 94% of residues in each homology model were located in the most favored regions (**S4 Fig**, **S5 Table**). Secondary structures of the wild-type and mutant S1PR1 proteins were analyzed and visualized using the STRIDE tool (**S5 Fig**). Notably, the coil region (T32–G33) of the native protein was converted into an α-helix in both the R120P and C184Y mutants. Similarly, the coil region (S248–E249) was transformed into an α-helix in the R120P and F125S mutants (**S5 Fig**). In the R120P mutant, an additional change was observed where the coil region (T109–K111) was converted into a 3–10 helix. In contrast, the F125S mutant did not exhibit further alterations compared to the native structure. The C184Y mutant displayed three significant secondary structure modifications: the 3–10 helix (74K–76F) was converted into a coil, the 3–10 helix (177L–180M) was altered into an α-helix at residue 177L and a coil at residues 178P–180M, and the 3–10 helix (188L–190S) was replaced by a coil (**S5 Fig**, **S5 Table**).

## Interatomic interaction and stability prediction

Interatomic interactions of the three selected mutant proteins were analyzed using the DynaMut2 server (**Fig 2**). The server predicted protein stability changes (ΔΔG) in kcal/mol, where positive values denoted stabilizing effects and negative values indicated destabilization. More negative ΔΔG values corresponded to greater destabilizing impacts on protein structure. The analysis revealed that the R120P (rs149198314) (**Fig 2A–2B**), F125S (rs1346744443) (**Fig 2C–2D**), and C184Y (rs1360246180) (**Fig 2E–2F**) mutants exerted destabilizing effects, with ΔΔG values of –0.43, –1.15, and –3.82 kcal/mol, respectively.

## Molecular docking and binding site prediction

Ligand-binding site analysis of the S1PR1 protein using CASTp 3.0 identified 46 key residues (**S6 Table**). Subsequent DynaMut2 analysis predicted three deleterious nsSNPs; however, only R120P and F125S were prioritized for molecular docking (**S6 Table**), as these substitutions occurred within the identified ligand-binding site. Molecular docking was performed for the wild-type S1PR1 and the R120P and F125S mutants against fingolimod-phosphate (FTY720-P), W146 (ML056), and the natural agonist S1P. Binding energies for all docked complexes are presented in **Table 1**. Docking results demonstrated significant differences between the wild-type and mutant proteins. FTY720-P exhibited strong binding to both the wild type (–8.4 kcal/mol) and the R120P mutant (–8.1 kcal/mol), but showed reduced affinity for the F125S mutant (–6.8 kcal/mol). W146 exhibited favorable binding across all receptor variants, with binding energies of –7.7 kcal/mol (wild type), –7.5 kcal/mol (R120P), and –7.4 kcal/mol (F125S), supporting its potential as a potent antagonist of the mutant receptors (**Fig 3**). As illustrated in **Fig 3A–3C**, S1P binding was weakened in the R120P and F125S mutants due to the loss of hydrogen bonds and π-interactions. Structural analysis (**Fig 4A–4C**) revealed that FTY720P maintained strong binding to the wild type and R120P but reduced affinity for F125S, reflecting impaired hydrophobic interaction. Interaction profiles (**Fig 5A–5C**) further demonstrated that W146 retained stable binding across all variants, with only minor reductions in affinity, underscoring its robustness as an antagonist. Collectively, these results suggested that the R120P and F125S mutations weakened S1P binding affinity, which may have impaired receptor activation.

## Dynamics simulation and protein–ligand complex stability

RMSD analysis was performed to assess the structural stability of wild-type and mutant S1PR1 protein–ligand complexes over a 200 ns MDS. The RMSD trajectories for the ligands S1P (CID: 5283560), W146 (CID: 6857802), and FTY720P (CID: 11452022) are presented in **Fig 6**. In the wild type, the S1P ligand exhibited relatively high fluctuations (6 Å),

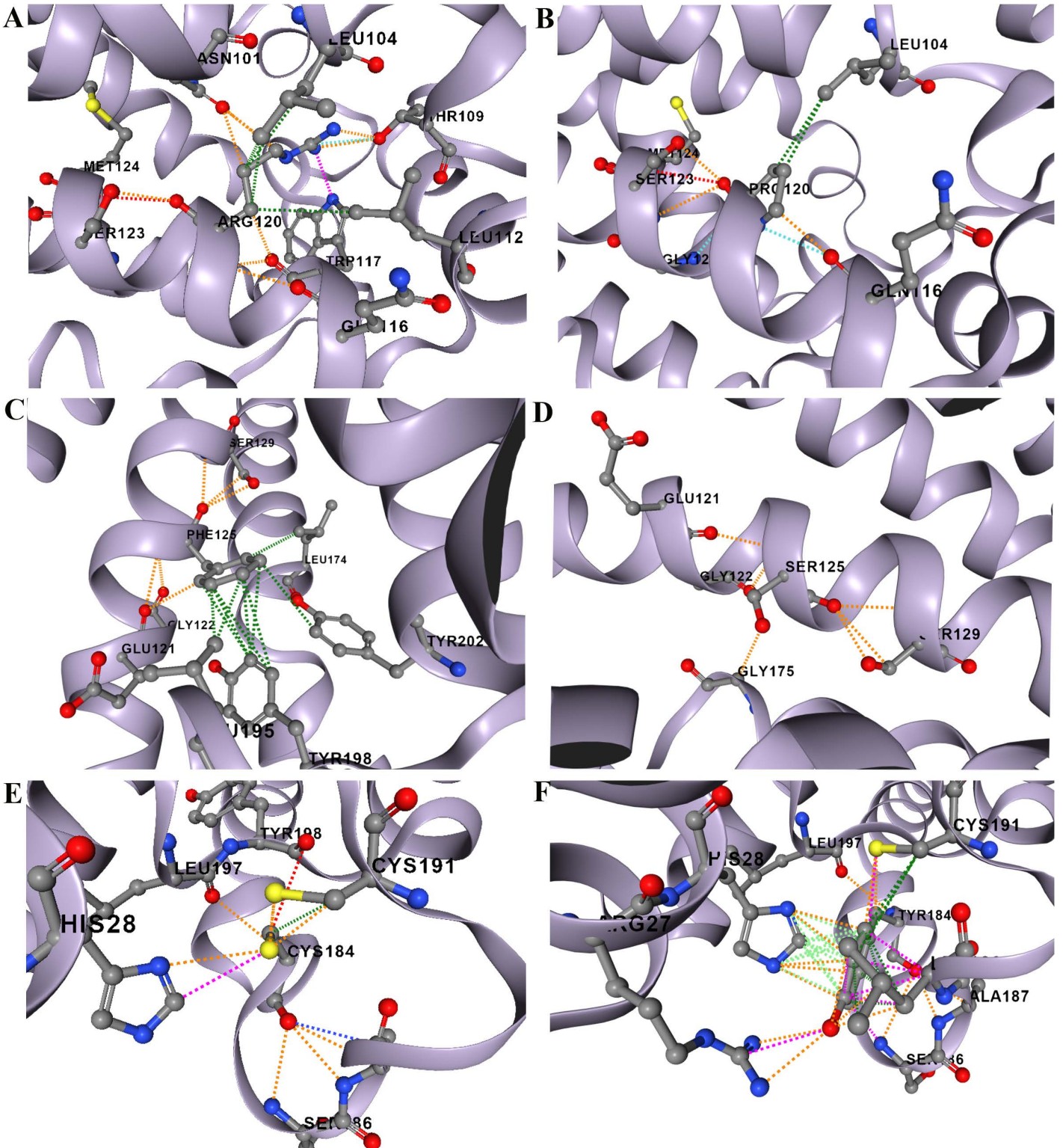

**Fig 2. Predicted changes in the interatomic interaction network of S1PR1 residues in wild-type and mutant variants. (A)** Wild-type R120, **(B)** Mutant R120P, **(C)** Wild-type F125, **(D)** Mutant F125S, **(E)** Wild-type C184, and **(F)** Mutant C184Y.

**Table 1. Binding affinity (kcal/mol) prediction of ligands with wild type and mutant structures of the human S1PR1 protein.**

| Receptor type | Binding affinity (kcal/mol) | | |
|---|---|---|---|
| | Sphingosine-1-phosphate (S1P) agonist | FTY720P agonist (Fingolimod-phosphate) | W146 antagonist |
| Wild protein (PDB ID_7VIF) | −7.1 | −8.4 | −7.7 |
| R120P | −6.8 | −8.1 | −7.5 |
| F125S | −6.4 | −6.8 | −7.4 |

whereas W146 and FTY720P displayed more stable profiles (3–4 Å). Introduction of mutations increased structural fluctuations, particularly for S1P (4–6 Å), suggesting destabilization of the complex. The R120P mutation further influenced protein dynamics; however, complexes with W146 and FTY720P remained comparatively stable (3–4.5 Å). Overall, these results indicated that the mutations altered the binding stability and dynamic behavior of the S1PR1–ligand complexes (**Fig 6**), potentially compromising ligand affinity and structural compatibility.

The RMSF analysis (**Fig 7**) was performed to examine residue-level flexibility in wild-type and mutant S1PR1 protein–ligand complexes during the 200 ns MD simulation. In the wild-type complexes, S1P (CID: 5283560) displayed high flexibility in loop regions, with peaks reaching 6.4 Å near residue indices 150–170 and 250–270, indicating enhanced mobility. W146 (CID: 6857802) and FTY720P (CID: 11452022) exhibited similar fluctuation patterns, with prominent peaks (6–7 Å) between residues 200–250, consistent with flexible loops contributing to ligand binding. In the F125S mutant, fluctuations in S1P- and W146-bound complexes were reduced, suggesting increased structural rigidity, whereas the FTY720P complex retained elevated flexibility (5.6 Å) near residues 220–250, indicating partial destabilization (**Fig 7**). Conversely, the R120P mutant induced distinct flexibility changes, particularly in the S1P complex, where peaks increased to 7.4 Å at residues 150–170 and 250–270, reflecting localized destabilization. Complexes with W146 and FTY720P exhibited moderate flexibility (5–6 Å) in similar regions, suggesting that the R120P substitution altered residue-specific dynamics, potentially weakening ligand binding interactions and overall stability (**Fig 7**).

## Binding interaction profiling of S1PR1 complexes

Interaction fraction analysis was performed to evaluate ligand-binding patterns in wild-type and mutant S1PR1 complexes (**Fig 8**). In the wild-type complexes, S1P (CID: 5283560), W146 (CID: 6857802), and FTY720P (CID: 11452022) formed strong interactions with residues such as Glu124, Ser128, and Val133, with interaction fractions reaching up to 1.75, indicating frequent and stable contacts. The F125S mutation led to a marked reduction in interaction strength for S1P and W146, particularly with Glu124 and Ser128, suggesting weakened binding stability. Conversely, FTY720P retained strong interactions near Glu124 and Tyr205, with interaction fractions close to 1.6, reflecting partially preserved binding affinity despite the mutation. The R120P mutation induced notable shifts in interaction patterns, especially for S1P, which formed new contacts with Leu150 and Gly175. In complexes with W146 and FTY720P, reduced interaction fractions and altered contact patterns indicated destabilization and rearrangement of the binding site (**Fig 8**). Collectively, these results highlight the residue-specific effects of mutations on ligand binding and protein–ligand interaction dynamics.

## Mutation-induced changes in binding interactions

Two-dimensional interaction diagrams (**Fig 9**) were used to visualize detailed binding interactions between the ligands S1P (CID: 5283560), W146 (CID: 6857802), and FTY720P (CID: 11452022) with wild-type and mutant S1PR1 proteins. In the wild-type complexes, S1P formed strong hydrogen bonds with Glu124, Ser128, and Arg120, stabilizing ligand binding. W146 exhibited extensive polar and hydrophobic interactions, particularly with Tyr78 and Arg120, enhancing its binding profile. FTY720P formed stable hydrogen bonds with Glu121 and multiple hydrophobic contacts, reinforcing its affinity (**Fig 9**).

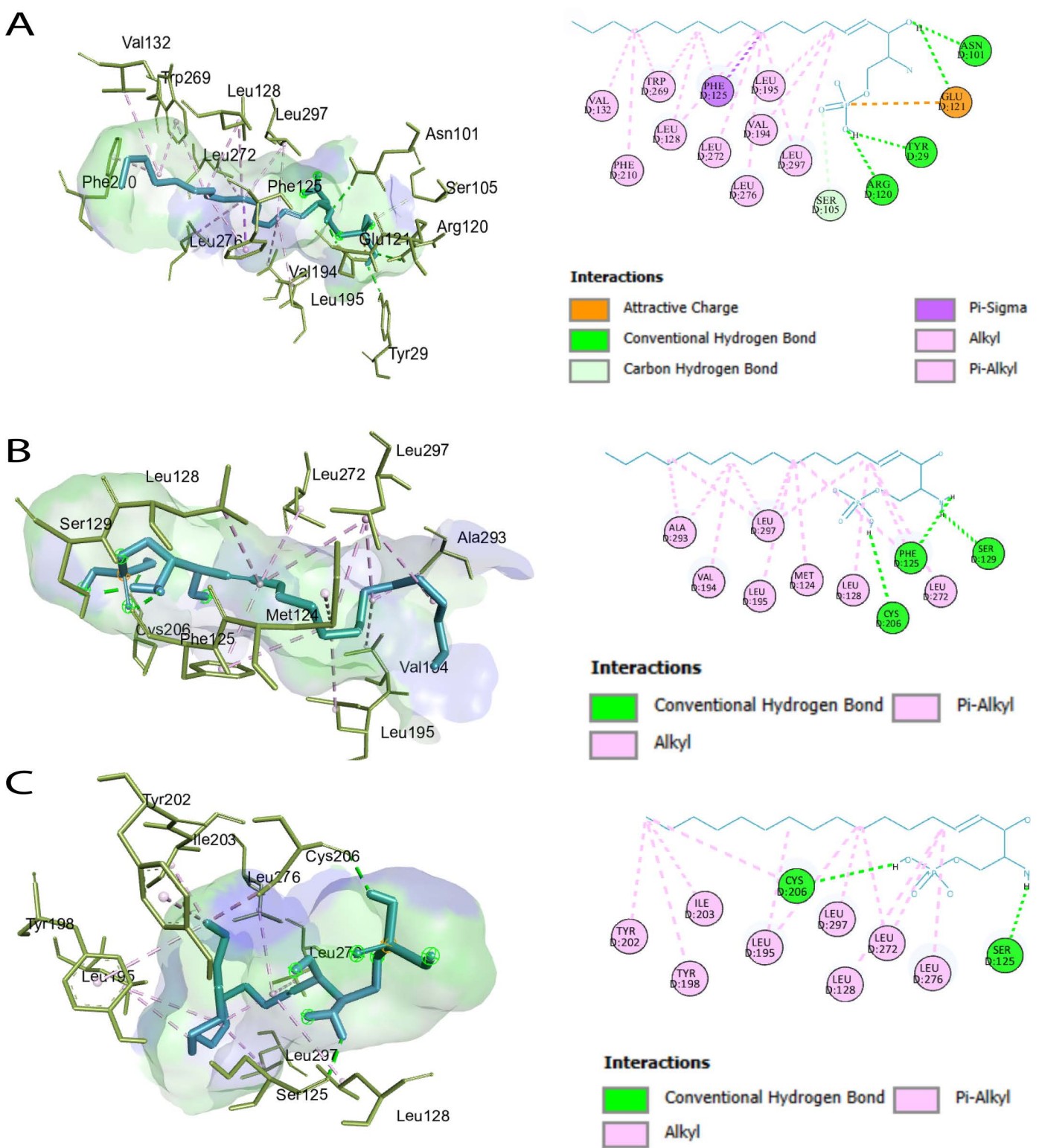

**Fig 3. Visualization of non-bonded interactions between S1PR1 residues and the natural agonist sphingosine-1-phosphate (S1P) ligand. (A)** Wild-type S1PR1 with S1P agonist, **(B)** R120P mutant with S1P agonist, and **(C)** F125S mutant with S1P agonist.

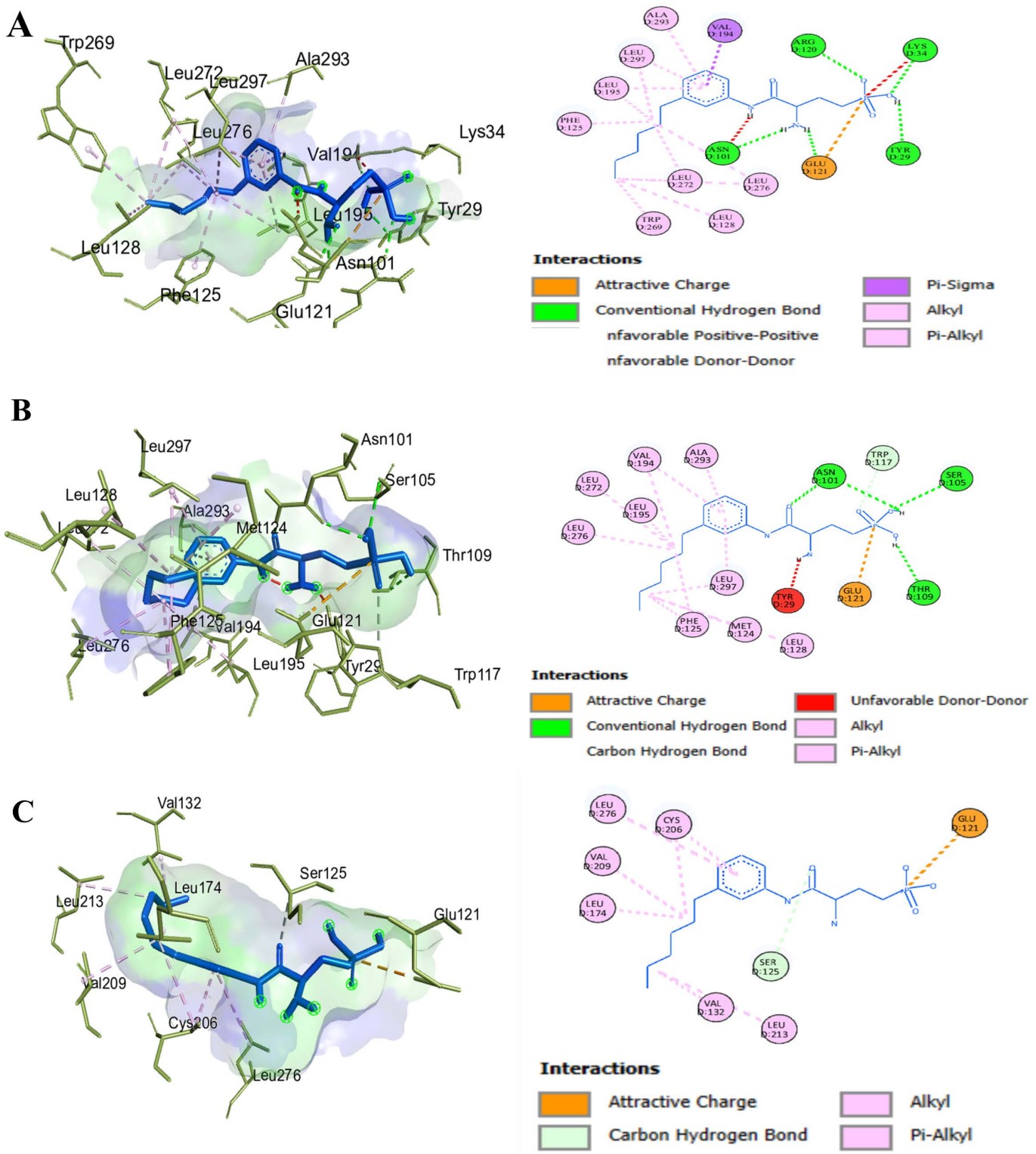

**Fig 4. Visualization of non-bonded interactions between S1PR1 residues and the FTY720P ligand. (A)** Wild-type S1PR1 with FTY720P agonist, **(B)** R120P mutant with FTY720P agonist, and **(C)** F125S mutant with FTY720P agonist.

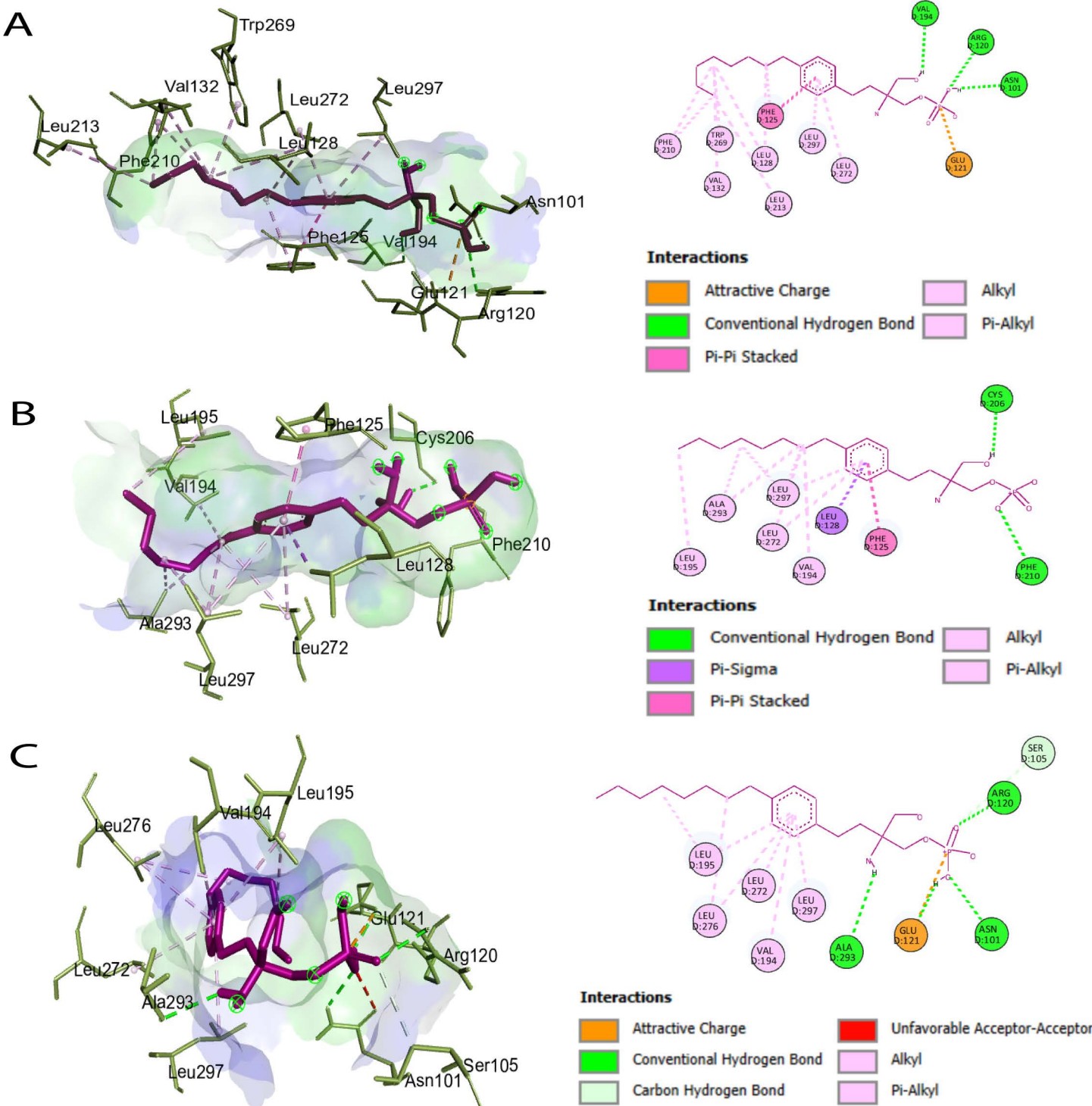

**Fig 5. Visualization of non-bonded interactions between S1PR1 residues and the W146 ligand. (A)** Wild-type S1PR1 with W146 agonist, **(B)** R120P mutant with W146 agonist, and **(C)** F125S mutant with W146 agonist.

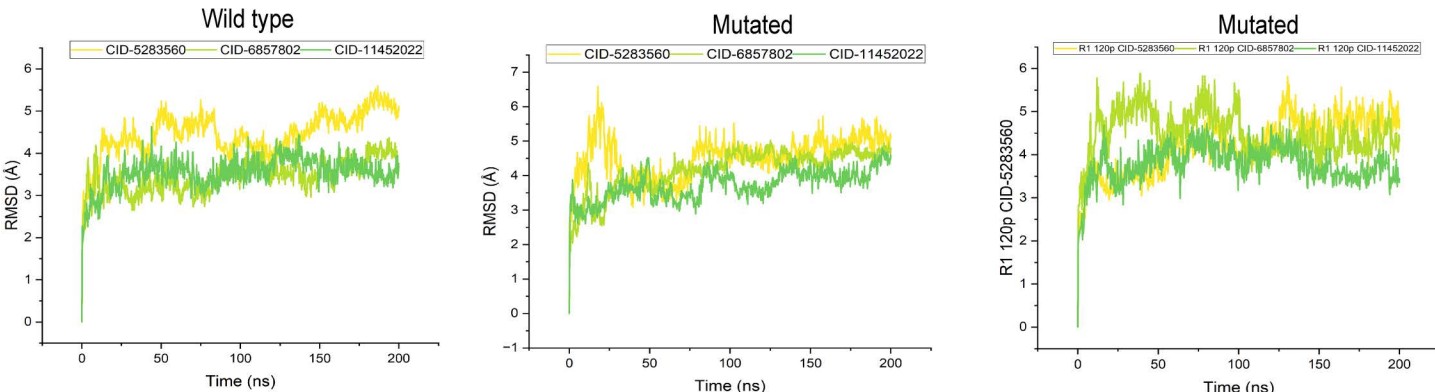

**Fig 6. Root Mean Square Deviation (RMSD) plots comparing wild-type (left), two mutated forms of S1PR1 protein including F125S (middle), and R120P (right) bound to S1P (CID: 5283560), W146 (CID: 6857802) and FTY720P (CID: 11452022) over a 200 ns simulation.** Each coloured line corresponds to a different ligand. Deviations in the RMSD profiles highlight changes in overall receptor stability and dynamics induced by specific mutations and ligand interactions.

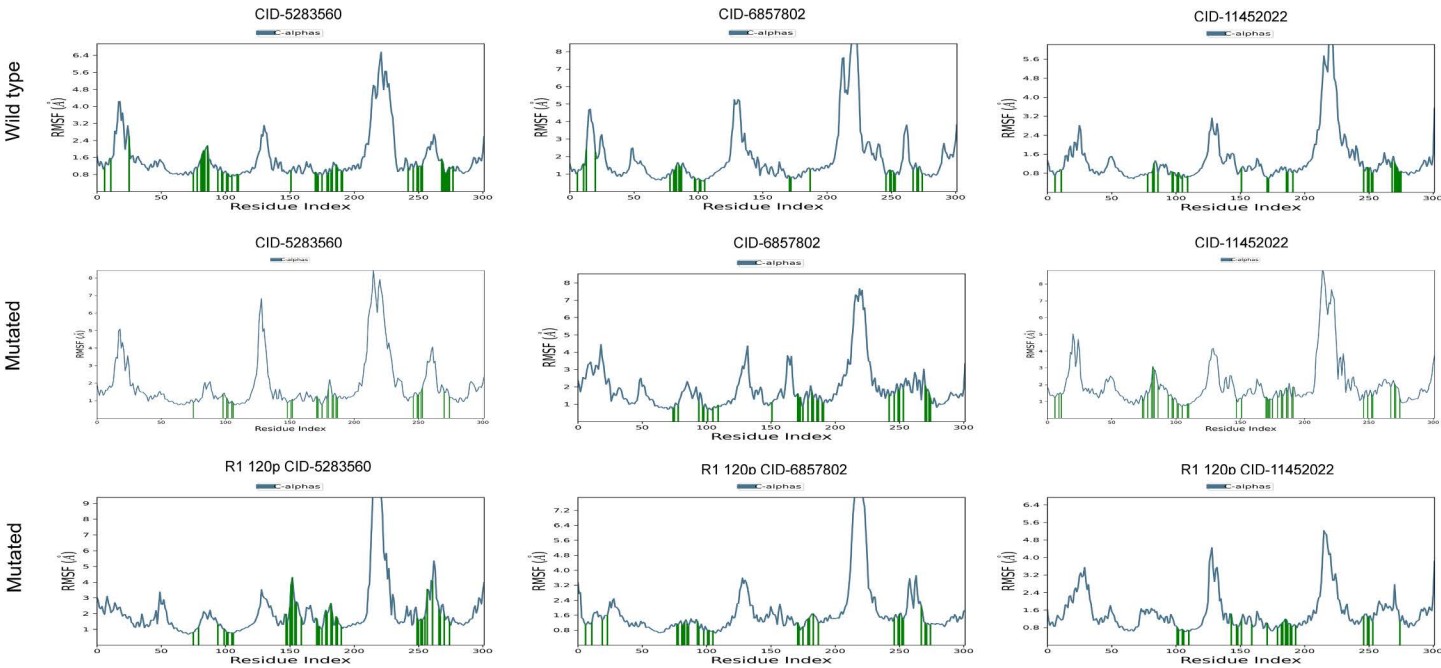

**Fig 7. Root Mean Square Fluctuation (RMSF) plots comparing wild-type (top row), two mutated forms of S1PR1 protein including F125S (middle row), and R120P (bottom row) bound to S1P (CID: 5283560), W146 (CID: 6857802) and FTY720P (CID: 11452022) ligands.** The black lines represent per-residue flexibility over the simulation, while green bars highlight fluctuations at specific regions. Shifts in RMSF peaks indicate how each mutation alters local protein dynamics in the presence of different ligands.

## Discussion

nsSNPs are genetic variations that result in amino acid substitutions and can significantly alter protein structure, stability, and function [67–69]. These alterations may affect protein–protein interactions, expression levels, folding, splicing, and ligand-binding capabilities, potentially contributing to disease mechanisms [27,69]. Traditional methods for SNP

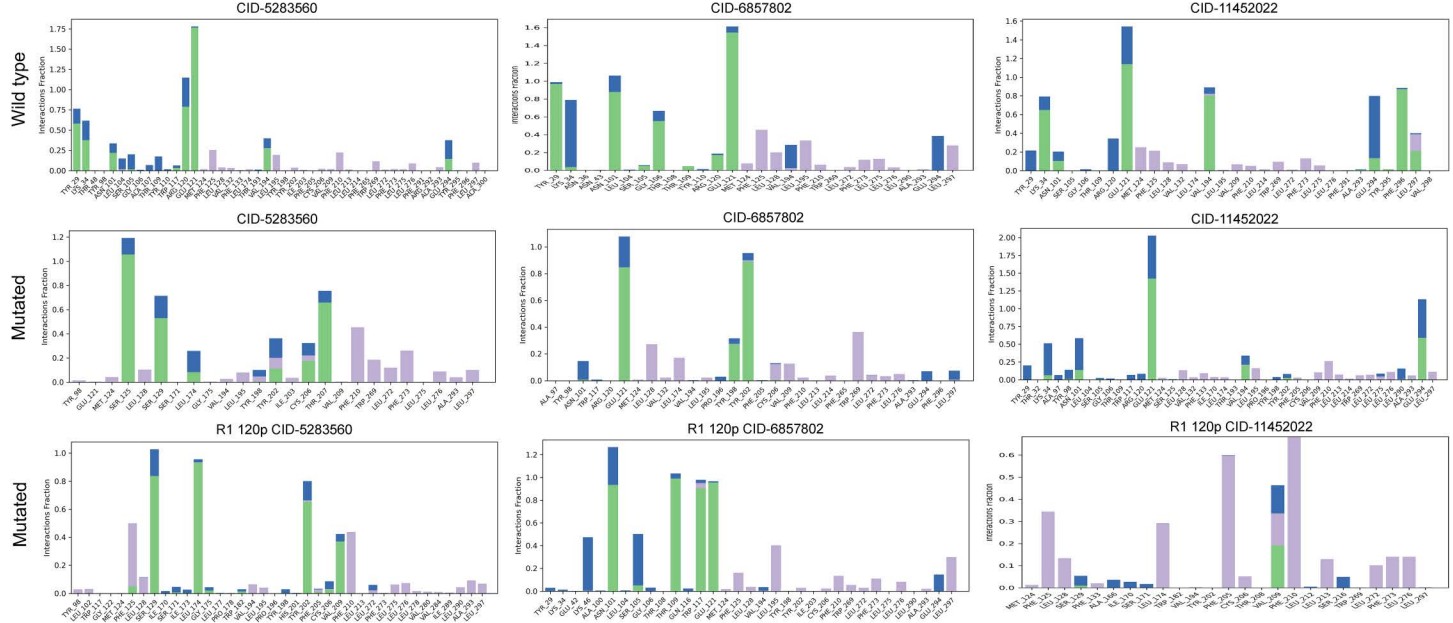

**Fig 8. Residue-wise interaction profiles of S1PR1–ligand complexes.** Bar charts comparing residue-wise interaction contributions in wild-type (top row), two mutated structures of S1PR1 protein including F125S (middle row), and R120P (bottom row) bound to S1P (CID: 5283560), W146 (CID: 6857802) and FTY720P (CID: 11452022) ligands. Each bar represents the magnitude of a specific interaction (e.g., electrostatic, hydrophobic) at each residue. The differences in bar height across conditions highlight how mutations and ligand identity influence the overall binding profile and stability of S1PR1-ligand complexes.

characterization, such as family-based linkage analysis and population-based association studies, are often costly and labor-intensive [70,71]. In contrast, computational approaches provide efficient and scalable means to analyze genetic variations and their phenotypic effects [72].This study focused on the *S1PR1* gene, which encodes a G protein-coupled receptor involved in immune regulation, angiogenesis, and cancer progression. Dysregulation of *S1PR1* gene has been implicated in multiple sclerosis and cancers including neuroblastoma, prostate, pancreatic, ovarian, and breast cancer [73,74]. By integrating multiple computational prediction algorithms and structural modelling approaches, this investigation prioritized S1PR1 variants most likely to exert harmful functional effects, providing candidates for experimental validation and potential therapeutic targeting.

Our comprehensive in-silico analysis identified five highly deleterious nsSNPs in the S1PR1 gene (R120P, F125S, C184Y, Y198C, and L275P) that are predicted to significantly compromise protein structure and function. The strategic location of all five variants within the 7-transmembrane (7-TM) domain is particularly noteworthy, as this region is critical for receptor conformation, ligand binding, and G protein coupling—essential functions for S1PR1-mediated signaling [75]. The conservation analysis provides compelling evidence for the functional importance of these residues. Three variants (R120P, F125S, C184Y) occur at highly conserved positions, suggesting these residues have been maintained throughout evolution due to critical structural or functional roles. The high phylogenetic similarity across primate orthologs further reinforces that S1PR1 has been under strong selective pressure, and mutations at these conserved sites are likely to be poorly tolerated. The differential conservation scores, with Y198C at moderately conserved and L275P at average conservation, suggest a gradient of functional constraint, though all five variants were consistently predicted as deleterious by multiple algorithms. An important distinction emerges from the structural context of these mutations. R120P was identified as an exposed functional residue, likely participating directly in ligand recognition or receptor activation, whereas F125S, C184Y, Y198C, and L275P are buried structural residues that probably maintain the protein's three-dimensional structure.

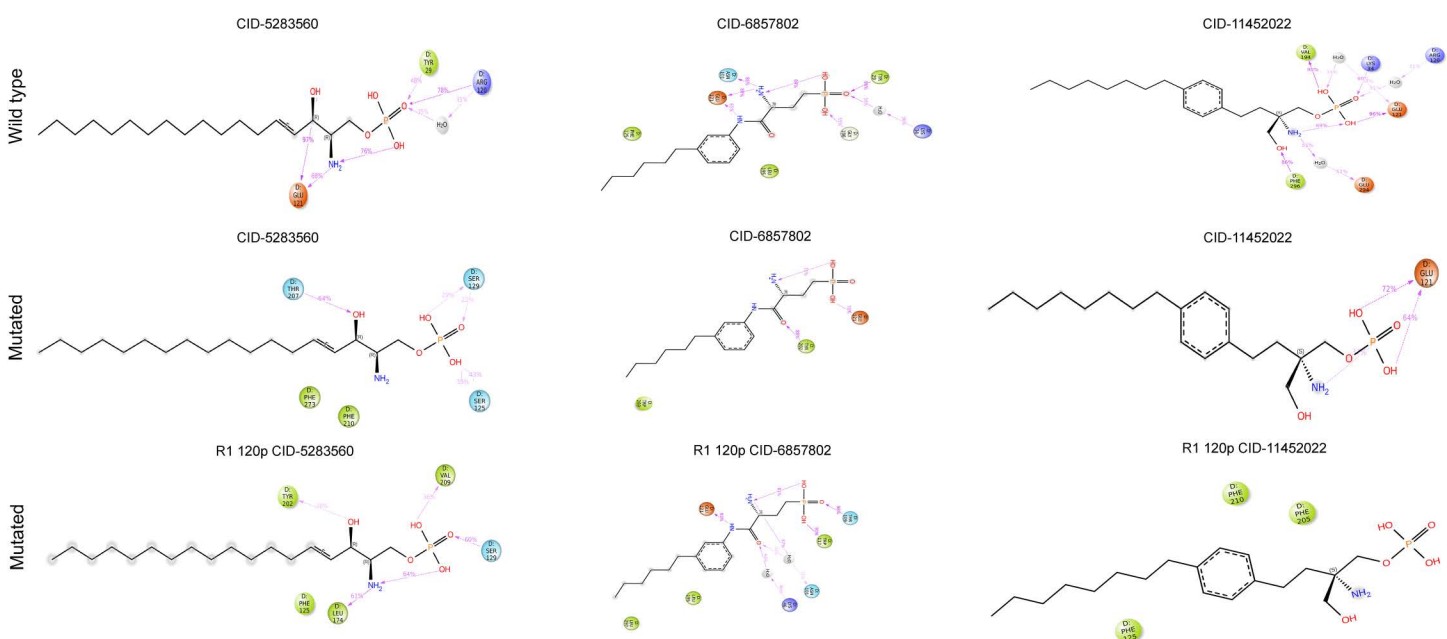

**Fig 9. Two-dimensional ligand interaction diagrams for wild-type (top row), two mutated structures of S1PR1 protein including F125S (middle row), and R120P (bottom row) bound to S1P (CID: 5283560), W146 (CID: 6857802) and FTY720P (CID: 11452022) ligands.** Colored spheres and lines represent distinct interaction types (e.g., hydrogen bonds, hydrophobic contacts), illustrating how specific mutations alter the binding environment and interaction patterns for each ligand.

This suggests these variants may impair S1PR1 function through different mechanisms: R120P potentially disrupting direct molecular interactions, while the buried variants may destabilize the overall fold or interfere with conformational transitions required for receptor activation. Human S1PR1 protein also showed close homology with *P. troglodytes* and *P. abelii*, reflecting recent common ancestry. These findings underscore the functional significance of S1PR1, suggesting that nsSNPs in conserved regions may have critical biological implications.

Structural and biochemical effects of the five key nsSNPs were further explored using Project HOPE. R120P and F125S involved increased amino acid size relative to the wild type, while C184Y showed decreased size. R120P altered amino acid charge (positive to neutral), potentially affecting protein-protein interactions [76]. Hydrophobicity changes were noted: F125S and C184Y decreased, while R120P increased, potentially disturbing essential hydrophobic interactions. Salt and cysteine bridges formed by R120P and F125S, respectively, indicated severe structural consequences, particularly as both mutations occur in the agonist-binding region. Structural changes in buried residues disrupt hydrogen bonds, disulfide bridges, and salt bridges, potentially causing thermodynamic instability, misfolding, and protein aggregation [77]. The observed secondary structure alterations across all mutants, coupled with predictions of increased flexibility and destabilization, suggest a significant compromise in protein stability. Furthermore, the conformational shift from buried to exposed in the C184Y mutant implies a potential impact on its functional interactions. R120P and F125S, located in the agonist-binding region, were docked with S1PR1 ligands *viz*. S1P, FTY720P (active form of Fingolimod for multiple sclerosis) [78,79], and antagonist W146 [53,80]. Their position suggests potential disruption of ligand binding and altered responsiveness to S1PR1-targeted therapies. Docking analysis revealed reduced binding affinity for both mutants compared to wild-type S1PR1. FTY720P and W146 exhibited stronger interactions with R120P, whereas W146 showed higher affinity for F125S, suggesting mutation-specific variations in ligand recognition.

MDS trajectory analysis further revealed mutation-specific effects on S1PR1-ligand complexes. R120P increased conformational fluctuations, particularly in S1P-bound complexes, with deviations up to 6 Å, suggesting destabilization [81]. F125S enhanced structural rigidity, reducing fluctuations and indicating a more stable ligand interaction [82]. RMSF analysis highlighted increased flexibility in key loop regions of R120P compared to wild-type and F125S [83]. Intramolecular interaction analysis demonstrated that R120P disrupted hydrogen bonds and hydrophobic interactions at key residues (e.g., Glu124, Ser128, Val133), while forming transient contacts with Leu150 and Gly175 [84]. F125S reduced S1P and W146 interaction strength, though FTY720P binding remained stable, particularly at Glu124 and Tyr205 [85]. Overall, R120P may impair ligand affinity and signalling, whereas F125S reduces receptor flexibility, illustrating how nsSNPs modulate ligand interactions and potentially influence drug response. Clinically, S1PR1 is involved in immune trafficking, angiogenesis, and cancer metastasis through pathways such as Ras/ERK, PI3K/AKT, and STAT3 [86,87]. Its overexpression in tumors and immune cells makes it a therapeutic target in multiple sclerosis and cancer [86,88]. nsSNPs like R120P (rs149198314), associated with cardiovascular disease and melanoma, along with unreported variants F125S and C184Y, may influence disease susceptibility and treatment response. Our findings suggest nsSNPs in S1PR1 protein may contribute to severe disease pathogenesis, including multiple sclerosis, breast, prostate, and ovarian cancers [86,88]. In this study, three deleterious SNPs namely R120P (rs149198314), F125S (rs1346744443), and C184Y (rs1360246180) were highlighted. Among these, R120P has been reported in European American populations and linked to cardiovascular disease [89], with one case associated with skin melanoma. F125S and C184Y are undocumented in current literature. While computational prediction methods provide functional predictions, limitations exist due to database incompleteness or errors. Experimental validation is necessary to confirm the biological and clinical significance of these variants, including effects on immune regulation, vascular function, and disease susceptibility. Additionally, non-coding SNPs affecting gene expression were not considered, representing a limitation, given the pivotal role of S1PR1 signalling in disease.

## Conclusion

This study provides comprehensive insights into the effects of nsSNPs in the human *S1PR1* gene, highlighting their structural, functional, and potential clinical implications. The deleterious mutations R120P and F125S were found to significantly influence protein stability and ligand-binding properties. Through integrated computational approaches including molecular docking and dynamics simulations (MDS), we demonstrated that these variants (*e.g.,* R120P and F125S) significantly alter S1PR1 protein stability and ligand-binding properties. The R120P mutation induced pronounced conformational fluctuations and disrupts critical ligand interactions, potentially impairing receptor function and downstream signalling. In contrast, F125S enhanced structural rigidity, stabilizing certain ligand interactions while reducing flexibility in other regions, indicating a nuanced effect on protein functionality. Collectively, these findings highlight the potential role of S1PR1 nsSNPs in modulating immune response and cancer pathways. Both of the predicted mutations may modulate S1PR1-mediated pathways involved in immune regulation, cancer progression, and other pathological processes. The in-silico predictions generated in this study provide a foundational resource for future experimental studies aimed at validating these mutations' effects on protein function and disease mechanisms. Furthermore, these results underscore the therapeutic potential of targeting S1PR1 in conditions such as multiple sclerosis and cancer. Future research should prioritize functional assays and structural biology techniques to confirm these predictions and support the development of mutation-specific drug design strategies.

## Supporting information

**S1 Table. Distribution of S1PR1 nsSNP (missense), synonymous, intron, in-frame deletions, in-frame insertions and others.**
(DOCX)

**S2 Table. Unanimous deleterious or damaging nsSNPs in the S1PR1 protein predicted by eight different tools.**
(DOCX)

**S3 Table. Structural impact prediction of S1PR1 high-risk pathogenic (deletorious) nsSNPs of S1PR1 protein.**
(DOCX)

**S4 Table. Evolutionary conservation prediction of S1PR1 protein through ConSurf.**
(DOCX)

**S5 Table. Protein model verification by PROCHECK and ERRAT.**
(DOCX)

**S6 Table. Forty-six key residues predicted in the S1PR1 protein.**
(DOCX)

**S1 Fig. Workflow for evaluating the functional and structural consequences of S1PR1 gene variants.** SNP data-sets were retrieved from dbSNP-NCBI, followed by functional impact assessment using multiple prediction tools (SIFT, PolyPhen-2, PROVEAN, etc.). Protein stability was analyzed with MUpro, I-Mutant 2.0, and NetSurfP-3.0, while domain identification was performed using Pfam. Evolutionary and conservation analyses were conducted with MEGA11, Iroki, and ConSurf. Homology modeling and model validation were carried out using SWISS-MODEL, PROCHECK, and ERRAT. Finally, molecular docking and molecular dynamics simulations were performed using PyRx and Schrödinger.
(DOCX)

**S2 Fig. Conservation profile of the human S1PR1 protein.** Analysis highlights three SNPs namely R120P, F125S, and C184Y as highly conserved positions predicted to exert the greatest impact on the structural integrity of the protein. These residues are marked within rectangular boxes for emphasis.
(DOCX)

**S3 Fig. Evolutionary phylogenetic analysis of the human *S1PR1* gene.** Human *S1PR1* gene shares a close evolutionary relationship with its homologs in *Pan troglodytes* and *Pongo abelii*.
(DOCX)

**S4 Fig. Ramachandran plots showing the backbone dihedral angle distributions of wild-type and mutant proteins, illustrating the conformational quality, stereochemical validity, and structural stability of the modeled protein structures.**
(DOCX)

**S5 Fig. Secondary structure analysis of wild type (A) and three selected mutants (B) R120P, (C) F125S and (D) C184Y by STRIDE program.** (E) Indicates the used legends of secondary structure icons. Changes in structural elements are shown in rectangular boxes.
(DOCX)

## Author contributions

**Conceptualization:** Sangram Biswas, Dipankar Sardar, Md. Arju Hossain, M. Nazmul Hoque.

**Data curation:** Sangram Biswas, Dipankar Sardar, Arif Hossain Ramjan, Ummah Kulsum Nazifa, Fatema-Tuz Zohora, Ishrat Jahan Esha, Chandrika Mondal, Abdul Barik.

**Formal analysis:** Sangram Biswas, Dipankar Sardar, Soharth Hasnat, Arif Hossain Ramjan, Ummah Kulsum Nazifa, Fatema-Tuz Zohora, Ishrat Jahan Esha, Chandrika Mondal, Abdul Barik.

**Investigation:** Sangram Biswas, Md. Arju Hossain, Soharth Hasnat, Arif Hossain Ramjan, Ummah Kulsum Nazifa, Fatema-Tuz Zohora, Ishrat Jahan Esha, Chandrika Mondal, Abdul Barik.

**Methodology:** Sangram Biswas, Dipankar Sardar, Soharth Hasnat, Arif Hossain Ramjan, Ummah Kulsum Nazifa, Fatema-Tuz Zohora, Ishrat Jahan Esha, Chandrika Mondal, Abdul Barik, M. Nazmul Hoque.

**Project administration:** Md. Arju Hossain, M. Nazmul Hoque.

**Resources:** Soharth Hasnat, Abdul Barik, M. Nazmul Hoque.

**Software:** Sangram Biswas, Soharth Hasnat, M. Nazmul Hoque.

**Supervision:** M. Nazmul Hoque.

**Validation:** Sangram Biswas, Md. Arju Hossain, Soharth Hasnat, M. Nazmul Hoque.

**Visualization:** Sangram Biswas, Dipankar Sardar, Md. Arju Hossain, Soharth Hasnat, Arif Hossain Ramjan, Abdul Barik.

**Writing – original draft:** Sangram Biswas, Dipankar Sardar, Soharth Hasnat, Arif Hossain Ramjan, Ummah Kulsum Nazifa, Fatema-Tuz Zohora, Ishrat Jahan Esha, Chandrika Mondal, Abdul Barik.

**Writing – review & editing:** Md. Arju Hossain, M. Nazmul Hoque.

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
