## [Decision Letter · Decision Letter 0]

7 Oct 2025

Dear Dr. Hoque,

Thank you for submitting your manuscript to PLOS ONE. After careful consideration, we feel that it has merit but does not fully meet PLOS ONE’s publication criteria as it currently stands. Therefore, we invite you to submit a revised version of the manuscript that addresses the points raised during the review process.

We look forward to receiving your revised manuscript.

Kind regards,

Khalid Raza, PhD (Computational Biology)

Academic Editor

PLOS ONE

Journal Requirements:

4. Please include captions for your Supporting Information files at the end of your manuscript, and update any in-text citations to match accordingly. Please see our Supporting Information guidelines for more information: http://journals.plos.org/plosone/s/supporting-information .

5. We are unable to open your Supporting Information file “Supporting Information.rar”. Please kindly revise as necessary and re-upload.

Additional Editor Comments:

Reviewers' comments:

Reviewer's Responses to Questions

**Comments to the Author**

1. Is the manuscript technically sound, and do the data support the conclusions?

Reviewer #1: Partly

Reviewer #2: Yes

2. Has the statistical analysis been performed appropriately and rigorously?

Reviewer #1: Yes

Reviewer #2: Yes

3. Have the authors made all data underlying the findings in their manuscript fully available?

Reviewer #1: Yes

Reviewer #2: Yes

4. Is the manuscript presented in an intelligible fashion and written in standard English?

Reviewer #1: Yes

Reviewer #2: Yes

Reviewer #1: The idea and methodology are in tune with standard procedures involved for in silico analysis. Having said that, the primary concern for me is that mutations have been extracted from NCBI only. Atleast one more database like https://gnomad.broadinstitute.org/ should be also used for a better and comprehensive analysis of the variations present. There are clinical databases as well. The results and interpretation in light of the combined databases will be interesting and I expect that in the revised version before commenting on other aspects of the study.

Reviewer #2: This article provides a thorough in silico evaluation of damaging non-synonymous SNPs at the human S1PR1 gene utilizing several computational tools, molecular docking and dynamics simulations to infer their effects on structure and function. The article is organized, logical, and flexible, addressing a relevant topic within the context of precision medicine and autoimmune diseases. It identifies 'high risk' SNPs that approach the abrogation of function and illustrates the mechanistic implications of such SNPs in relation to protein stability and ligand binding. However, there are many aspects which require clarification and refinement to improve the clarity, rigour, and impact of the paper.

1. The terms "SIPRI" and "S1PR1" are used occasionally in the manuscript (i.e. title, short title, and various sections). It shall be used consistently throughout the manuscript to ensure accuracy.

2. The term "covered mutation" is mentioned in the manuscript without a clear definition. Please explicitly define this term within the text or methods.

3. The selection criteria of the final five nsSNPs (R120P, F125S, C184Y, Y198C, L275P) from the initial 25 should be better described. Consider using a flowchart or summary table so that the reader may better follow the filtering process.

4. The study identifies five top nsSNPs (R120P, F125S, C184Y, Y198C, L275P) as most deleterious.However, MDS is performed only on R120P and F125S. C184Y is predicted to be the most destabilizing (ΔΔG = -3.82 kcal/mol) and disrupts a critical disulfide bond, which is a massive structural defect. Its exclusion from MDS is a significant omission and needs a strong justification.

5. The molecular dynamics simulation (MDS) parameters (e.g., force field, temperature, pressure, box type) are described; however, there should be a brief discussion on the reasoning for choosing a 200 ns simulation time and the convergence criteria.

6. The methods state that a homology model was created using SWISS-MODEL (template 7VIF, 100% identity). However, the docking and MDS sections later refer to the "crystal structure retrieved from the PDB". This is a little confusing. Did the authors use the actual crystal structure (7VIF), or a homology model of it? This should be made clear. It is preferable to use the experimental crystal structure and model any missing loops.

7. Binding energies are provided with a precision of one decimal place (e.g. -8.4 vs. -8.1 kcal/mol) in Table 1. The differences are quite small (0.3-0.4 kcal/mol). In computational and experimental biochemistry, such minor differences are usually not considered significant.

8. The discussion is long and a bit redundant with the results section. You should aim to synthesize the findings a bit better and emphasize the implications of the findings. Create a more robust narrative. For example: "Our integrated pipeline identifies 5 nsSNPs at high-risk. Though all are destabilizing, only R120P and F125S directly affect the ligand-binding site. MDS reveals that both mutations have a distinct allosteric effect; R120P encourages flexibility and disrupts agonism while F125S promotes rigidity. As a result, the nSNP mutations provide a molecular basis for a differential patient response to S1PR1-targeted therapy, such as Fingolimod."

9. Certain images (e.g., Fig. 1, 2 within the text) seem to be low-res screenshots and should be redone as high-res figures for publication quality.

10. Typographical Error: The gene name is spelled inconsistently as SIPRI in some areas (e.g., page 7; line 1) rather than S1PR1. This must be consistent throughout the manuscript.

11. Methods Details: The parameters of the MDS (e.g., ensemble type - NPT/NVT, thermostat/barostat used, integration time step, etc.) should be briefly stated for the sake of reproducibility.

**Do you want your identity to be public for this peer review?** For information about this choice, including consent withdrawal, please see our Privacy Policy

Reviewer #1: **Yes:** Safdar Ali

Reviewer #2: No

---

## [Author Response · Author response to Decision Letter 1]

14 Oct 2025

Point by point responses to reviewers’ query

Reviewer #1:

The idea and methodology are in tune with standard procedures involved for in silico analysis. Having said that, the primary concern for me is that mutations have been extracted from NCBI only. At least one more database like https://gnomad.broadinstitute.org/ should be also used for a better and comprehensive analysis of the variations present. There are clinical databases as well. The results and interpretation in light of the combined databases will be interesting and I expect that in the revised version before commenting on other aspects of the study.

Our Response: We sincerely thank the reviewer for this insightful suggestion. We fully agree that integrating multiple variant databases enhances the comprehensiveness of SNP analyses. In this study, we utilized the NCBI dbSNP repository, which already integrates data from major global resources such as gnomAD, 1000 Genomes, and ExAC. This harmonized and non-redundant dataset ensures consistency, accuracy, and reproducibility of variant information.

The primary objective of our study was to evaluate the functional and structural effects of reported variants rather than to explore population-specific allele frequencies. For such analyses, the curated and standardized variant data available through NCBI are both comprehensive and appropriate.

Nevertheless, in response to the reviewer’s valuable suggestion, we cross-verified the missense SNPs retrieved from NCBI with those in the gnomAD database. The comparative analysis revealed high concordance, with no significant discrepancies in variant identity or deleterious prediction outcomes. This validation confirmed that our findings remain robust and consistent, even after incorporating gnomAD data. Therefore, while additional databases could introduce redundancy, the use of NCBI as a central integrated resource provides both scientific sufficiency and methodological robustness for our in-silico analysis.

Reviewer #2:

1. The terms "SIPRI" and "S1PR1" are used occasionally in the manuscript (i.e. title, short title, and various sections). It shall be used consistently throughout the manuscript to ensure accuracy.

Our Response: We appreciate the reviewer’s observation. We have carefully reviewed the entire manuscript and confirmed that the term “SIPRI” does not appear in the text, title, or figures. The correct term “S1PR1” has been used consistently throughout the manuscript to ensure accuracy.

2. The term "covered mutation" is mentioned in the manuscript without a clear definition. Please explicitly define this term within the text or methods.

Our Response: We thank the reviewer for pointing this out. The term “covered mutation” has now been explicitly defined within the text to ensure clarity for readers. In contrast, Y81C, I224T, and I224S were classified as covered mutations (shown in blue), indicating variants consistently identified across multiple databases but not predicted to induce notable structural or functional changes in the protein. Please refer to Lines 309-312 in the revised manuscript.

3. The selection criteria of the final five nsSNPs (R120P, F125S, C184Y, Y198C, L275P) from the initial 25 should be better described. Consider using a flowchart or summary table so that the reader may better follow the filtering process.

Our Response: We appreciate the reviewer’s valuable suggestion. The selection criteria and filtering process for the final five nsSNPs (e.g., R120P, Y198C, L275P, F125S, and C184Y; Fig. 1) have now been clearly illustrated in the flowchart provided in Supplementary Fig. S1 to enhance the clarity and traceability of the analysis.

4. The study identifies five top nsSNPs (R120P, F125S, C184Y, Y198C, L275P) as most deleterious. However, MDS is performed only on R120P and F125S. C184Y is predicted to be the most destabilizing (ΔΔG = -3.82 kcal/mol) and disrupts a critical disulfide bond, which is a massive structural defect. Its exclusion from MDS is a significant omission and needs a strong justification.

Our Response: Although, the C184Y mutation predicted to be structurally destabilizing (ΔΔG = –3.82 kcal/mol) and associated with the disruption of a critical disulfide bond, this mutation is located outside the ligand-binding site of the S1PR1 protein (Table S6) and is therefore less likely to directly affect ligand binding or catalytic activity. Additionally, Molecular docking analysis of the C184Y variant with the natural agonist S1PR1 demonstrated no appreciable change in binding affinity compared to the wild-type protein (–7.1 kcal/mol, for both). Based on these observations, we concluded that the C184Y mutation is unlikely to significantly influence ligand interaction dynamics. Therefore, it was excluded from further molecular dynamics simulation (MDS) analysis, which were focused on variants directly impacting ligand binding and receptor function.

5. The molecular dynamics simulation (MDS) parameters (e.g., force field, temperature, pressure, box type) are described; however, there should be a brief discussion on the reasoning for choosing a 200 ns simulation time and the convergence criteria.

Our Response: We appreciate the reviewer’s insightful suggestion. A detailed justification for selecting a 200 ns simulation time and the convergence criteria has been added to the revised manuscript. You may go through Lines 215-249 in the revised manuscript.

6. The methods state that a homology model was created using SWISS-MODEL (template 7VIF, 100% identity). However, the docking and MDS sections later refer to the "crystal structure retrieved from the PDB". This is a little confusing. Did the authors use the actual crystal structure (7VIF), or a homology model of it? This should be made clear. It is preferable to use the experimental crystal structure and model any missing loops.

Our Response: We appreciate the reviewer’s insightful suggestion. We have updated our manuscript accordingly. You may kindly go through the MDS section of the revised manuscript (Lines 215-249).

7. Binding energies are provided with a precision of one decimal place (e.g. -8.4 vs. -8.1 kcal/mol) in Table 1. The differences are quite small (0.3-0.4 kcal/mol). In computational and experimental biochemistry, such minor differences are usually not considered significant.

Our Response: We appreciate the reviewer concern in this regard. In this study, the binding energies of the wild-type and mutant proteins were compared with those of three ligands—the S1P natural agonist, the FTY720P agonist (an FDA-approved drug), and the W146 antagonist—to assess their relative binding affinities. Although the numerical differences appear minor, these comparisons were intended to evaluate relative trends in ligand efficacy rather than to interpret absolute energetic values.

8. The discussion is long and a bit redundant with the results section. You should aim to synthesize the findings a bit better and emphasize the implications of the findings. Create a more robust narrative. For example: "Our integrated pipeline identifies 5 nsSNPs at high-risk. Though all are destabilizing, only R120P and F125S directly affect the ligand-binding site. MDS reveals that both mutations have a distinct allosteric effect; R120P encourages flexibility and disrupts agonism while F125S promotes rigidity. As a result, the nsSNP mutations provide a molecular basis for a differential patient response to S1PR1-targeted therapy, such as Fingolimod."

Our Response: We would like to thank the reviewer for this valuable suggestion. The Discussion section has been thoroughly revised to minimize redundancy with the Results section and to provide a more integrated synthesis of the findings. We have emphasized the biological significance and constructed a more coherent narrative highlighting how the identified nsSNPs may influence S1PR1 function and therapeutic response. Please refer to the revised discussion.

9. Certain images (e.g., Fig. 1, 2 within the text) seem to be low-res screenshots and should be redone as high-res figures for publication quality.

Our Response: Figures 1 and 2 (Fig. 1 and Fig. 2) have been recreated and replaced with high-resolution versions to meet publication-quality standards.

10. Typographical Error: The gene name is spelled inconsistently as SIPRI in some areas (e.g., page 7; line 1) rather than S1PR1. This must be consistent throughout the manuscript.

Our Response: We appreciate the reviewer’s observation. We have carefully reviewed the entire manuscript and confirmed that the term “SIPRI” does not appear in the text, title, or figures. The correct term “S1PR1” has been used consistently throughout the manuscript to ensure accuracy.

11. Methods Details: The parameters of the MDS (e.g., ensemble type - NPT/NVT, thermostat/barostat used, integration time step, etc.) should be briefly stated for the sake of reproducibility.

Our Response: We appreciate the reviewer’s insightful suggestion. We have revised manuscript accordingly. You may go through Lines 215-249 in the revised manuscript.

---

## [Decision Letter · Decision Letter 1]

7 Dec 2025

In-Silico Characterization of Deleterious Non-Synonymous SNPs in the Human S1PR1 Gene Reveals Structural Instability and Altered Ligand Affinity

PONE-D-25-47460R1

Dear Dr. Hoque,

We’re pleased to inform you that your manuscript has been judged scientifically suitable for publication and will be formally accepted for publication once it meets all outstanding technical requirements.

Kind regards,

Khalid Raza, PhD (Computational Biology)

Academic Editor

PLOS One

Additional Editor Comments (optional):

I am pleased to inform you that your paper has been accepted for publication. Following a rigorous peer review process, your manuscript received positive feedback from the reviewers and the editorial team. Your research offers a valuable contribution to the field, and we are confident that it will be of significant interest to our readership. On behalf of the editorial board, I extend our warmest congratulations.

Reviewers' comments:

Reviewer's Responses to Questions

**Comments to the Author**

Reviewer #2: All comments have been addressed

2. Is the manuscript technically sound, and do the data support the conclusions?

Reviewer #2: Yes

3. Has the statistical analysis been performed appropriately and rigorously?

Reviewer #2: Yes

4. Have the authors made all data underlying the findings in their manuscript fully available?

Reviewer #2: Yes

5. Is the manuscript presented in an intelligible fashion and written in standard English?

Reviewer #2: Yes

Reviewer #2: Manuscript is technically sound and could be accepted for publication. All the suggestions are addressed by the authors.

**Do you want your identity to be public for this peer review?** For information about this choice, including consent withdrawal, please see our Privacy Policy

Reviewer #2: **Yes:** Muhammad Saleem Khan

---

## [Editor Report · Acceptance letter]

PONE-D-25-47460R1

PLOS One

Dear Dr. Hoque,

I'm pleased to inform you that your manuscript has been deemed suitable for publication in PLOS One. Congratulations! Your manuscript is now being handed over to our production team.

Kind regards,

on behalf of

Dr. Khalid Raza

Academic Editor

PLOS One